# Apparatus for Dry Deposition of Aerosols on Snow

Nicholas D. Beres[1,2] and Hans Moosmüller[1]

[1]Laboratory for Aerosol Science, Spectroscopy, and Optics, Desert Research Institute, Reno, NV, 89512, United States
[2]University of Nevada-Reno, Reno, NV, 89512, United States

*Correspondence to*: Nicholas D. Beres (Nic.Beres@DRI.edu)

**Abstract.** Deposition of light absorbing aerosol on snow can drastically change the albedo of the snow surface and the energy balance of the snowpack. To study these important effects experimentally and to compare with theory, it is desirable to have an apparatus for such deposition experiments. Here, we describe a simple apparatus to generate and evenly deposit light absorbing aerosols onto a flat snow surface. Aerosols are produced (combustion aerosols) or entrained (mineral dust aerosols) and continuously transported into a deposition chamber placed on the snow surface where they deposit onto and into the snowpack, thereby modifying its surface reflectance and albedo. We demonstrate field operation of this apparatus by generating black and brown carbon combustion aerosols and entraining hematite mineral dust aerosol and depositing them on the snowpack. Changes in spectral snow reflectance is demonstrated qualitatively through pictures of snow surfaces after aerosol deposition and quantitatively by measuring hemispherical-conical reflectance spectra for the deposited areas and for adjacent snowpack in its natural state. Additional potential applications for this apparatus are mentioned and briefly discussed.

## 1 Introduction

Aerosols in the Earth-atmosphere system play a critical role in radiative forcing and climate change (IPCC, 2013). However, our understanding of how they affect the cryosphere upon deposition onto snow surfaces is still limited (Qian et al., 2015), particularly for aerosols other than black carbon (Skiles et al., 2018). Understanding aerosol-cryosphere interactions is important on several levels, including: (1) the radiative properties of the snowpack modified by deposited aerosols (Warren and Wiscombe, 1980; Warren 1982; Gardner and Sharp, 2010; Bond et al., 2013; Qian et al., 2015), which alter the seasonal timing of snow melt, runoff, and water management (Painter et al., 2010; Dozier, 2011), (2) the relation between deposited nutrients and the snowpack biosphere (Thomas and Duval, 1995; Jones, 1999; Kuhn, 2001; Hodson et al., 2008), and (3) snowpack chemistry and photochemistry, which are influenced by the deposited chemical compounds and their interaction with solar radiation (Grannas et al, 2007). One limitation in our understanding is establishing links between theoretical and experimental results because it is difficult to experimentally characterize these interactions in a controlled manner; generally, aerosol deposition on snow is spatially and temporally very inhomogeneous, and often deposition and its immediate effects are minor. One scarcely-used but effective experimental method is to artificially deposit aerosols of interest directly onto the snow surface.

Atmospheric deposition is an important process by which an exchange of nutrients, gasses, and particles takes place between the atmosphere and land and sea surfaces. Many deposition processes are irreversible; for example, once the deposition of particles occurs, the probability of re-entrainment is low. Atmospheric constituents are removed from the atmosphere through dry and wet deposition (Seinfeld and Pandis, 2016). Dry deposition is facilitated by gravitational settling, inertial impaction, and Brownian diffusion processes. Wet deposition, on the other hand, involves the scavenging of gasses and particles by clouds and precipitation through dissolution, CCN activity, and collision processes. Aerosol and gasses can enter into the ice-grain matrix of the snowpack through different means and alter chemical, physical, and radiative properties (Kuhn, 2001; Grannas et al., 2007). For snowpack radiative processes and energy balance, the deposition of light-absorbing aerosols (i.e., black carbon (BC), brown carbon (BrC), and mineral dust, Moosmüller et al., 2009) is of special interest because deposition of even minute quantities of strongly light-absorbing aerosols drastically increases the co-albedo of the snow surface in the visible and near-visible spectral regions, where pure snow is "snow-white" with hardly any intrinsic absorption (Warren, 1982).

Carbonaceous aerosols in the atmosphere, including BC and BrC (Chakrabarty et al., 2010; Lack et al., 2014), are dominantly generated by incomplete combustion of fossil and biomass fuels with significant additional generation of secondary organic aerosols through oxidation of volatile precursors in the atmosphere (Bond et al., 2004; Lin et al., 2014). These aerosols are lofted into the atmosphere, where, during transport of a few days to weeks, they undergo secondary processing (Jimenez et al., 2009) and eventually are removed from the atmosphere through wet or dry deposition (Bond et al., 2013). BC is a ubiquitous light-absorbing aerosol in the atmosphere that directly affects Earth's radiative budget (Jacobson, 2001; Bond et al., 2013) and the cryosphere (Hegg et al., 2009; Flanner et al., 2009; Hadley and Kirchstetter, 2012; Qian et al., 2015). Recently, BrC has become of interest regarding its role in atmospheric light absorption (Laskin et al., 2015) and even more recently a topic of concern for affecting snow albedo and energy balance (Dang and Hegg, 2014; Doherty et al., 2014; Lin et al., 2014; Wu et al., 2016). The fraction of BC versus BrC mass emitted by combustion sources depends greatly on a number of factors, including fuel type, fuel moisture content, and packing density (Sumlin et al., 2018), combustion phase (Patterson and McMahon, 1984; Reid et al., 2004; Bond et al., 2004), and other elements of the system (Chen and Moosmüller, 2006). For BC, the imaginary part $\kappa$ of the refractive index is relatively large ($\kappa \approx 0.79$) and varies little over the visible and near-visible wavelength regions (Bond et al., 2006; Moosmüller et al., 2009). Because the imaginary part of the refractive index is so high for BC relative to that of snow, a small amount of BC present in the snowpack significantly increases the co-albedo in those spectral regions (Warren and Wiscombe, 1980; Cereceda-Balic et al., 2018). For BrC, the imaginary part of the refractive index generally increases towards shorter visible and ultra-violet wavelengths (Chakrabarty et al., 2010; Moosmüller et al, 2009; Moosmüller et al., 2011). Combustion systems – typically smoldering wildland fires – that produce a significant amount of BrC will have smoke plumes that are brownish or yellowish in appearance because of this spectral dependence (Andreae and Gelencsér, 2006). For this study, the boreal peat fuel used is the same as used by Sumlin et al. (2018), who found a value of $\kappa = 0.014$ at 375 nm, $\kappa = 0.003$ at 532 nm, and $\kappa = 0.002$ at 1047 nm. We are not aware of any published studies that explore the impact of deposition from a BrC dominated emission source on snow or identify the spectral reflectance signature of BrC particles deposited onto snow.

In addition to the importance of carbonaceous aerosol deposition on snow, mineral dust deposition has been shown to be an important driver of early snowmelt in some mountains (e.g., Painter et al., 2007; Skiles et al., 2012; Painter et al., 2012; Painter et al., 2017). The light absorption of mineral dust is mostly caused by iron oxides, such as hematite ($Fe_2O_3$) (Moosmüller et al., 2012; Zhang et al., 2015; Engelbrecht et al., 2016). Hematite is a mineral dust component with global-scale abundance and has been found to significantly absorb light in the ultra-violet and visible wavelengths - the imaginary refractive index of hematite is larger the shorter visible wavelengths and near UV ($\kappa \approx 0.90$ at 460 nm) and decreases with increasing wavelength (e.g., $\kappa \approx 0.035$ at 2500 nm) (Querry, 1985; Moosmüller et al, 2009; Dubovik et al, 2002) - and may have the largest optically absorptive impact of all mineral dust components found in the cryosphere (Hegg et al, 2010). Mineral dust deposition on snow is a major concern for the albedo feedback effects in certain areas of the cryosphere (McConnell et al., 2007; Painter et al., 2007; Skiles et al., 2012; Zhou et al., 2017), and accurate experimental representation of dust depositions is critical to understanding their interaction with snow.

Previous experiments have utilized aerosol artificial deposition techniques to test non-natural snow surface albedo perturbations, but with varying success. Conway et al., (1996) manually mixed soot and volcanic ash with loose snow in a bucket before being spread over a large plot. These methods were useful in understanding the characteristics and vertical location of aerosol during conditions of melt, but they likely restructured the bulk aerosol into agglomerations larger than what can be seen from normal deposition. Brandt et al. (2011) measured the snow albedo resulting from mixing commercially-available soot into tap water and spraying the mixture through a commercially-available snow-making machine over a field of artificial snowpack, produced with the same means. More recently, Peltoniemi et al. (2015) distributed chimney soot, glaciated silt, and volcanic sand onto the snow surface using a "salt shaker" in an attempt to measure the bidirectional reflectance factor of the resulting, contaminated snow. In addition, the snow albedo response to absorbing impurities on snow has been characterized by Singh et al. (2010) "spraying…soil equally on the surface".

Our work presented here describes a simple apparatus to evenly deposit aerosols in an artificial manner onto a flat snow surface through dry deposition for the study of snow-aerosol interactions. Tests during the snow season of 2015-2016 were conducted at the Cold Regions Research and Engineering Laboratory/UC Santa Barbara Energy Site (CUES, Bair et al, 2015) on Mammoth Mountain, California, USA. During the 2016-2017 and 2017-2018 snow seasons, experiments were conducted at Tamarack Lake in the Carson Range of the Sierra Nevada in Nevada, USA. Controlled deposition experiments using the apparatus for hematite mineral dust entrainment and combustion aerosol production of BC and BrC to modify snow surface reflectance are presented as examples.

## 2 Description and Operation of Apparatus

The deposition apparatus presented here is composed of two primary components: the aerosol production or entrainment chamber and the deposition chamber. The materials used for the air source and production or entrainment chamber depend on the type of test aerosol generated or entrained and deposited onto the snow surface. The deposition chamber is the same for any type of aerosol used.

### 2.1 Aerosol Production Chamber (Combustion Aerosols)

A schematic diagram of the aerosol production chamber for generating combustion aerosols is shown in Figure 1. It consists of a flat plywood base with an area of ~0.5 m$^2$ and a ~10 L near-cylindrical galvanized steel volume fitted with a ~1 L volume cone to its top; the cone funnels aerosols towards the chamber outlet. Inside the chamber, a ~6 cm high stage is mounted on the plywood base, directly above a perforated inlet that disperses inlet air horizontally under the combustion stage. The bottom rim of the chamber is lined with a rubber seal that, when mounted to the plywood base with clamps, becomes near-airtight. The combustion stage can be adapted to house different combusting configurations to fit the user's needs; generally, it symmetrically distributes inlet air around the stage and serves as platform for fuels to be burned. The air source used in the combustion aerosol configuration has three purposes: (1) to provide the appropriate amount of air for (incomplete) combustion to take place, (2) to provide continuous flow to move the combustion aerosols vertically (with assistance from flame buoyancy) through the production chamber towards the outlet, and to create a pressure differential between the production chamber and the deposition chamber that ultimately moves aerosols from the production chamber to the deposition chamber and finally onto and into the snowpack. If the heat from the combustion source is too intense, a small amount of metal tubing arranged in a coil may be necessary to cool the combustion air before being transported through any conducting rubber tubing, as shown in Figure 3.

The air source used for this configuration is a battery-powered, 12-volt air pump that provides inlet air into the entrainment chamber, with a flow rate of ~5.3 LPM at the outlet of the production chamber. This outlet is connected to the deposition chamber mostly via 6 mm OD copper tubing, as rubber or plastic tubing near the production chamber may melt from the combustion heat.

The user of this apparatus can vary the mass of combustion material deposited by limiting the amount of time the fuel is combusting. Here, we purposefully produced heavy depositions for optical inspection and verification of the proper operation of the apparatus.

### 2.2 Aerosol Entrainment Chamber (Mineral Dust Aerosols)

For pulverized, dry solids such as mineral dusts, the entrainment chamber consists of a 1 L volume glass Büchner flask (e.g., Jensen, 2006), but used in an opposing manner, providing positive pressure rather than a vacuum. Here, the flask is filled with a generous amount (~90 g) of the dust of interest, changing this amount allows for controlling the dust mass deposited onto the snow in the deposition chamber. Compressed air is introduced into the flask through the large opening at its top, sealed with a rubber stopper. The flask's short, horizontal tube acts as the outlet for entrained aerosols and is connected directly to the deposition chamber via conductive tubing. Dry solids are adequately re-suspended using short, intense bursts of air typically between 200 and 300 kPa in pressure from a hand-operated, positive-displacement piston pump, such as a common bike floor pump. A similar setup for the entrainment of mineral dust has previously been used by Moosmüller et al. (2012), Engelbrecht et al. (2016), and Piedra et al. (2018).

### 2.3 Deposition Chamber

The deposition chamber for all aerosols is a near-cylindrical galvanized steel volume that measures approximately 50 cm in diameter at its midpoint and has a volume of ~100 L. This piece of the apparatus is shown in Figure 2. There is an inlet at the top of the volume that allows for aerosol flow from the production or entrainment chamber and disperses the aerosols into the volume by splitting the aerosol flow into three separate outflows symmetrical about the center of the chamber, for uniform aerosol deposition onto the snow surface. There is no exhaust or outlet for air entering the deposition chamber besides through the snowpack, and the flowrate of air entering the deposition chamber may vary. The design of the instrument is to pump

aerosol into the snow, and the depth of particle capture by the snowpack depends on a variety of issues, as discussed in Kuhn (2001) and references therein.

Previous implementations of the apparatus included a 90-mm diameter, 12V DC-powered fan that was thought to help facilitate the dispersion of particles within the deposition chamber. This battery-powered fan was mounted in the
approximate centroid of the chamber and was tested in several different flow directions. The left and center panels of Figure 4 represent depositions of hematite while using this fan in the deposition chamber to test for increased deposition uniformity. However, the presence of the fan created less-uniform depositions, so the fan was removed. The right panel of Figure 4 shows a deposition of hematite with the most up-to-date configuration of the deposition apparatus, without fan.

## 3 Characterization of Apparatus and Results

To demonstrate the effectiveness of this deposition apparatus, it was tested during the spring of the 2016, 2017, and 2018. Deposition of three different aerosol types was demonstrated: BC and BrC in the combustion aerosol production configuration and a dry, sieved mineral dust, hematite ($Fe_2O_3$), to test the entrainment configuration. Information regarding the size distribution of each tested aerosol can be found in the Supplemental Data accompanying this manuscript. Hematite deposition was tested at the Cold Regions Research and Engineering Laboratory/UC Santa Barbara Energy Site (CUES, Bair et al, 2015)
on Mammoth Mountain, California (39° 38' 34.8" N, 119° 1' 44.3994" W, 2940 masl) on 11 May 2016. BrC and BC deposition was tested in the Carson Range of the Sierra Nevada Mountains at Tamarack Lake, NV (39° 19' 2.517" N, 119° 54' 19.512" W, 2694 masl) on 02 May 2017 and 24 April 2018, respectively; the lake was frozen and covered by snow deposited on its ice surface, creating a flat, homogeneous natural snowpack that is ideal for testing this apparatus.

To quantify the effect of aerosol deposition onto a natural snow surface, both subjective and objective measures were
used. The uniformity of the aerosol deposition was inspected visually for aerosol types that are visually dark, or optically absorbing, as for the hematite deposition in Figure 4. In addition, the hemispherical-conical reflectance factor (HCRF, Schaepman-Strub et al., 2006) was measured with an Analytical Spectral Devices (ASD) FieldSpec3 spectroradiometer to verify that aerosol was indeed altering the surface spectral reflectivity of the snow and to spectrally quantify that effect. Note that the measured HCRF is related to the surface albedo, controlling solar energy input to the snowpack (Schaepman-Strub et
al., 2006). After deposition of the different aerosol species, HCRF was measured for the deposition area as well as for an adjacent, untreated, natural surface that mirrored the snow properties of the deposition area prior to the experiment. This allowed the characterization and verification of the snow reflectivity reduction due to aerosol deposition. Ten measurements of HCRF were performed and averaged for each deposition area. This averaged value is presented throughout this paper with one standard deviation of the mean.

Measurement information and environmental conditions are summarized in Table 1. Solar zenith and azimuth angles have been obtained from the date and time of the measurement using the NOAA ESRL Solar Position Calculator (https://www.esrl.noaa.gov/gmd/grad/solcalc/azel.html). The spectrometer optical input is a cone from the instrument's bare optical fiber with full-angle field of view of 25°; all observation angles are nadir within a few degrees. Meteorological information was obtained for the Tamarack Lake site from the nearby NRCS SNOTEL site at Mt Rose (2683 masl,
https://wcc.sc.egov.usda.gov/nwcc/site?sitenum=652) and for the CUES site from its meteorological instrumentation (http://snow.ucsb.edu/).

### 3.1 Example: Combustion Aerosol Deposition

### 3.1.1 Black Carbon (BC) Deposition

Generation and deposition of BC aerosol took place during the spring of 2018 at Tamarack Lake. A kerosene lamp was used
to produce BC aerosol (Arnott et al., 2000; Arnold et al., 2014) to be deposited onto the snow surface. First, the 12-volt pump was started to begin moving air through the complete apparatus and provide enough air to sustain the combustion; without this forced air, the flame would quickly be extinguished due to lack of oxidant. The kerosene lamp was filled with fuel and lit before being placed onto the combustion stage and the production chamber being sealed with clamps. Then, the deposition chamber was lowered onto the desired snow surface, thereby initiating BC deposition on the snow. For this experiment, the
kerosene lamp was ignited, and aerosol emissions were pumped into the deposition chamber for 45 minutes. After this time, the flame was extinguished manually, and air was pumped through the apparatus for an additional 15 minutes to facilitate

further deposition of aerosol onto and into the snowpack. The deposition chamber was promptly removed, and spectral reflection properties of the deposited and nearby unsoiled snow surfaces were characterized with HCRF measurements.

This deposition of BC aerosol reduced the HCRF from ~0.87 ± 0.003 to ~0.58 ± 0.006 at 500 nm, an approximate reduction of ~33% corresponding to a large increase (~factor of 3) of the solar energy input to the snowpack. An image of the deposition area is presented in Figure 5 along with corresponding HRCF measurements made for the deposition area and for adjacent natural snow. Of the example depositions presented, this deposition was the least uniform.

### 3.1.2 Brown Carbon (BrC) Deposition

The fuel combusted for BrC aerosol generation consisted of boreal peat samples collected from interior Alaska, USA. Details of this fuel – including its collection and preparation – have been given by Chakrabarty et al. (2016). The optical, physical, and chemical properties of aerosol emissions from combustion of this fuel have been extensively studied to evaluate the impact of its combustion emissions on air quality and radiative forcing in the atmosphere through optical, physical, and chemical characterization (Chakrabarty et al., 2016; Samburova et al., 2016; Sengupta et al., 2018; Sumlin et al., 2017; Sumlin et al., 2018). Prior to combustion, the fuel samples were placed into a round, insulated container to mimic simple, real-world conditions in which there is little lateral heat flux due to largely homogeneous horizontal conditions. The fuel samples were burning with nearly exclusively smoldering phase combustion, producing OC-rich biomass burning aerosols. The fuel samples were smoldering for ~30-40 minutes; after this period, we continued to pump air through the apparatus for an additional 15 minutes to facilitate further deposition of aerosol onto and into the snowpack. Following the deposition, the deposition chamber was removed and HCRF measurements were made on the deposited and nearby undisturbed snow surfaces.

BrC deposited onto the snow surface greatly reduced the measured HCRF, but only at the shorter visible and UV wavelengths. The effect presented here reduced the reflectivity of a springtime snowpack from 0.92 ± 0.003 for the adjacent natural snowpack to 0.53 ± 0.02 at 350 nm, a reduction of ~39% for the BrC deposited snow. The resulting image and spectrum are presented in Figure 6. The image of the deposition clearly shows a yellowish appearance, an indication of BrC's preferential absorption of blue wavelengths. This visual impression and the corresponding spectrum (Figure 6), confirms the dominance of BrC over BC aerosol within the snowpack, as BC would have reduced snow reflectivity across the visible and near-visible spectrum and resulted in a greyish or blackish appearance on the snow surface.

### 3.2 Example: Deposition of Re-suspended Mineral Dust - Hematite

Mineral dust in the atmosphere – and that deposited into the cryosphere – has varying optical properties depending on its chemical and mineralogical composition. Here, synthetically-made, pure hematite ($Fe_2O_3$) was used as surrogate for strongly absorbing, red-colored mineral dust with high iron-oxide and hematite content. Depositions of hematite were made on a relatively level, undisturbed snow surface at the CUES research site.

A generous amount – approximately 90 g – of hematite dust was placed into the Büchner flask and was entrained and pumped into the deposition chamber as described in Section 2. The dust was allowed to gravitationally settle onto the snow surface for approximately 30 minutes, at which time the deposition chamber was removed and HCRF measurements were made for the deposited snow surface and the adjacent natural snow. The resulting image and spectrum from this test are shown in Figure 7. The resulting HCRF was reduced by approximately 35% in the 350 – 575-nm wavelength range when compared to that of the nearby natural snowpack.

### 3.3 Challenges and Further Development

The apparatus described here is not perfect but a work in progress and will benefit from further development by us and others. Some of the limitations and potential biases are outlined below.

A substantial but not quantified fraction of the aerosols generated in the production or entrainment chamber is lost during transport, deposited on walls and tubing as evidenced by the surfaces of the apparatus darkening and acquiring a typical smell for BrC depositions. Additionally, the authors made no effort to monitor the mass of deposited material or how deep the aerosol penetrated into the snowpack; instead, they leave this issue for future development. Particle deposition onto the snow is not perfectly homogenous and this homogeneity varies from deposition to deposition. The dominant factor controlling homogeneity of the deposit seems to be wind, which causes the deposition to favor the lee side of the deposition area due to the air's ability to permeate and travel through the snowpack (e.g. Waddington et al., 1996). Additionally, this apparatus may

alter the grain size of snow located at the top of the snowpack due to heating of the air inside the deposition chamber. Monitoring the grain size of surrounding natural snow and comparing to that of grains within the deposition area after the experiment can shed light on the induced effects. By conducting this analysis using the methods outlined in Nolin and Dozier (2000), we have concluded that, for this set of experiments, there is no consistent change in grain size. Temperature artifacts
in the deposition chamber could partly be mitigated by painting the exterior surface white to better reflect incident solar radiation.

       Ensuring identical conditions inside and outside of the deposition chamber would be very challenging. Additional instrumentation within the deposition chamber can help quantify the impact that outside air temperature, incoming solar radiation, outgoing thermal infrared radiation, and wind speed could have on experimentation. Perhaps, the use of an identical
deposition chamber, one with and one without introduced aerosols could minimize this problem.

## 4 Conclusion and Discussion

The apparatus described here provides a means to generate or entrain and to artificially deposit aerosols evenly onto a snow surface. The apparatus has been proven to efficiently deposit carbonaceous aerosols – BC and BrC – from two combustion sources as well as entrained dry mineral dust onto snow, thereby altering the surface reflectivity of snow. The reduction of
spectral surface reflectivity was verified by measuring the directional surface reflectance within the area of deposition and comparing it to the reflectance of neighboring natural snow. To the authors' knowledge, this study is the first to deposit primary aerosol from combustion sources in-situ, which provides the user of the apparatus with a novel tool to investigate the impact that these prolific snow impurities have after deposition. This investigation has proven that future applications of this apparatus are numerous.

20        The type of aerosol being deposited, the total mass of that aerosol, and the environmental conditions surrounding the deposition area can be adjusted by the users to suit their needs. The methods outlined by Skiles et al. (2016) to retrieve the refractive index of deposited aerosols from directional reflectance measurements can be applied to the artificial deposition methods described here with some additional radiative transfer analysis. Similarly, one can apply this apparatus to the testing of snow radiative transfer codes (e.g., SNICAR, Flanner and Zender, 2005; TARTES, Libois et al., 2013) as to their treatment
of the influence of impurities deposited onto the snowpack. Beres et al., (2019, in preparation) deposit varying concentrations of BrC onto the snow surface and verify measured total organic carbon concentrations to their albedo-reducing effect in an aerosol-snow coupled radiative transfer model.

       The growing importance of understanding the link between radiative forcing by a variety of aerosols found in the atmosphere and their relationship to the change in physical, chemical, and optical properties of snow and ice are a field that
can be utilized by this apparatus in the future. BrC aerosol have an impact on the cryosphere that is still being understood based on the close proximity of BrC-rich fuel sources and the proclivity of wildfire present in the Boreal forests of the northern latitudes (Flannigan et al., 2009; Oris et al., 2014; Beres et al., 2019, in prep.). Additionally, the increased emissions of dusts across the globe (Mahowald et al., 2010) and their impact on snow radiative forcing can benefit from this device. For example, one could entrain other globally important dusts, such as those found in Engelbrecht et al. (2016), using this apparatus. While
the first implementation of any new apparatus is imperfect, the usefulness and interest to the cryosphere sciences can benefit from this work.

*Data Availability.* The authors provide the numerical values of HCRF spectra shown in figures within the manuscript in the
supplementary material. Additionally, the authors have provided normalized, laboratory-measured size distributions that correspond to aerosols produced for this study under similar conditions.

*Competing Interests.* The authors declare that they have no conflict of interest.

*Acknowledgements.* This material has been supported in part by NASA EPSCoR under Cooperative Agreement No. NNX14AN24A, NASA ROSES under Grant No. NNX15AI48G, and by the National Science Foundation under Grant No. AGS-1544425. It is a pleasure to acknowledge Deep Sengupta for help with field experiments, Adam Watts for supplying the peat samples, and Jeff Dozier, Ned Bair, and Mammoth Mountain Ski Resort for generously providing access to and assistance at CUES.

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

**Table 1:** Overview of deposition and HCRF collection information

| Site | Deposited Aerosol | Altitude (m) | Date of Experiment | SZ* (°) | **AZ**** (°) | Avg. air temp.*** (°C) |
|---|---|---|---|---|---|---|
| CUES, CA, USA | Hematite ($Fe_2O_3$) | 2940 | 10 May 2016 | 21.5 | 154.3 | 8.4 |
| Tamarack Lake, NV, USA | Brown carbon (BrC) | 2694 | 02 May 2017 | 34.6 | 235.0 | 10.4 |
| Tamarack Lake, NV, USA | Black Carbon (BC) | 2694 | 24 April 2018 | 26.7 | 167.6 | 11.2 |

\*       SZ: solar zenith angle at time of HCRF data collection

\*\*      AZ: solar azimuth angle at time of HCRF data collection

\*\*\*     Average temperature during the HCRF data collection period. For the Tamarack Lake site, values are taken from the
5   Mt Rose NRCS SNOTEL site, which lies 0.9 km southeast at 2683 m in altitude. For the CUES site, these are taken directly
from the CUES meteorological instrumentation.

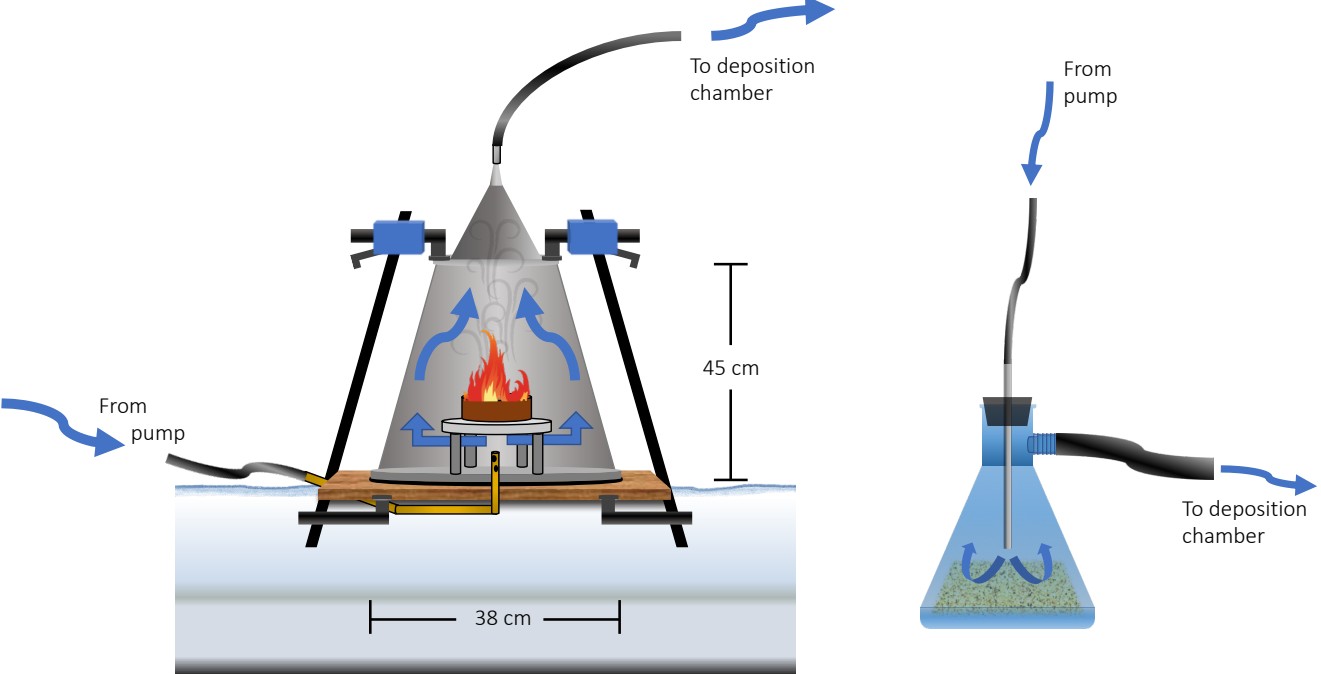

**Figure 1:** A battery-powered pump provided combustion and transport air into the aerosol production chamber for combustion aerosol (left), which flows to the deposition chamber at a flow rate of ~5.3 LPM. A 1-liter glass Buchner flask was used as the entrainment chamber for dry, pulverized dusts (right), which was provided short, quick bursts of air from a positive-displacement piston pump. The schematics are not drawn to scale.

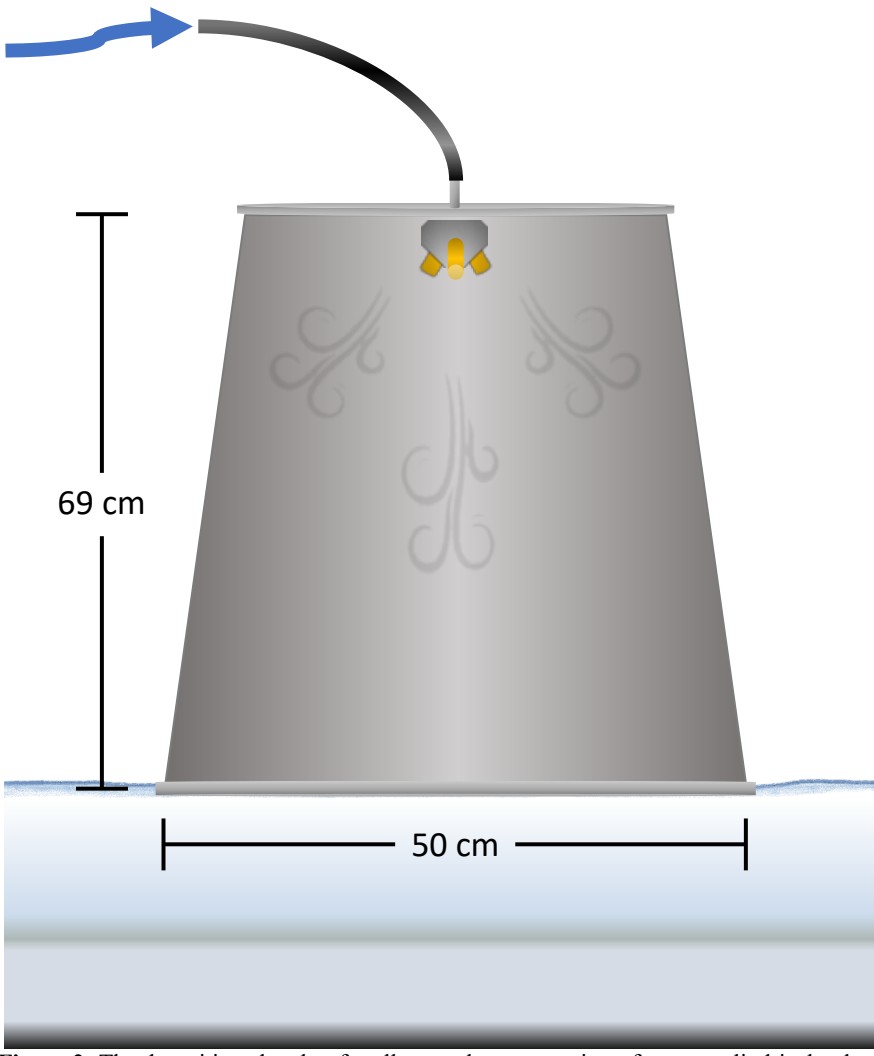

**Figure 2:** The deposition chamber for all aerosol types consists of a near cylindrical volume with aerosols pumped into the inlet at the top of the chamber from the production or entrainment chamber. The deposited area measures approximately 0.79 m².

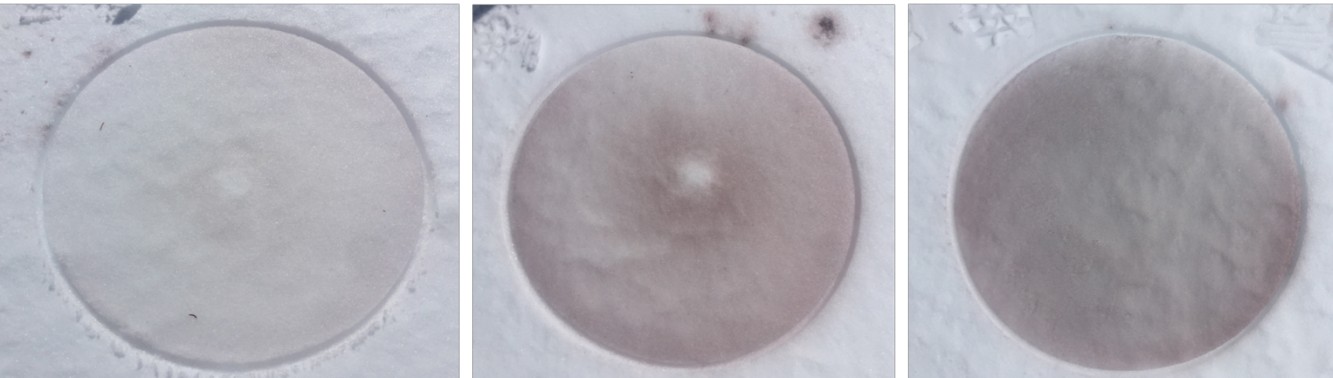

**Figure 3:** Use of the apparatus at the Tamarack Lake site for depositing BC aerosol.

**Figure 4:** Images of three hematite depositions allow for visual inspection of the deposition uniformity. The left and center panels represent a previous apparatus design which featured a fan mounted inside the deposition chamber, which created less-uniform depositions. The right panel represents a hematite deposition using the final configuration of the apparatus.

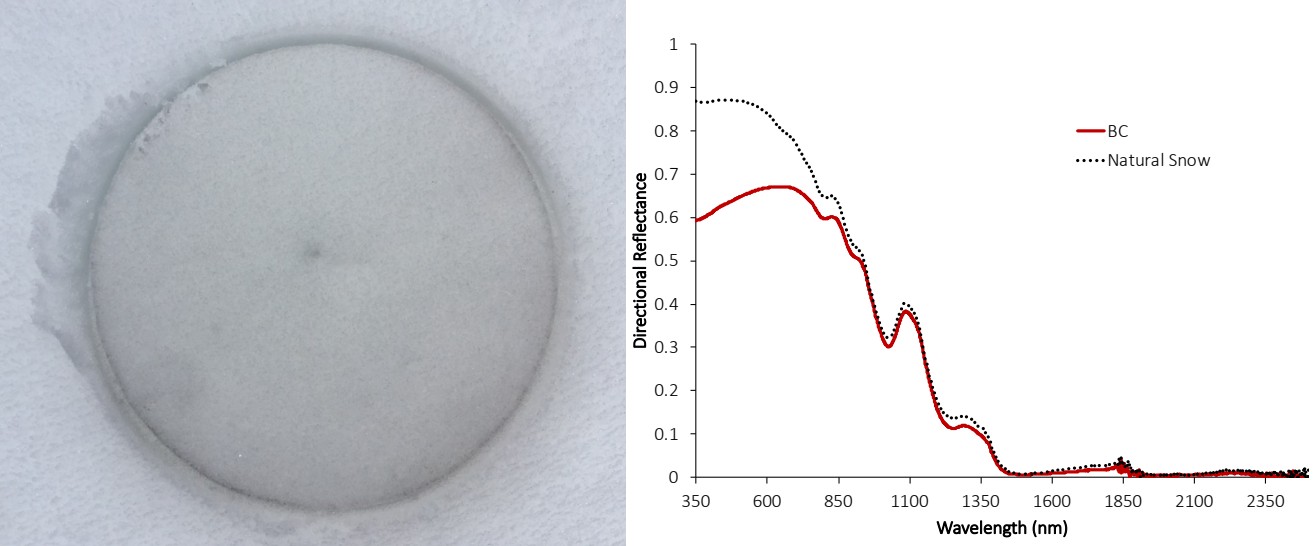

**Figure 5:** Image and HCRF spectra for a BC deposition and adjacent natural snow in May of 2018 at the Tamarack Lake site. This BC deposition drastically reduced the high natural snow reflectivity in the visible and near-visible spectral regions. Solar zenith: 21.5°; solar azimuth: 154.3°.

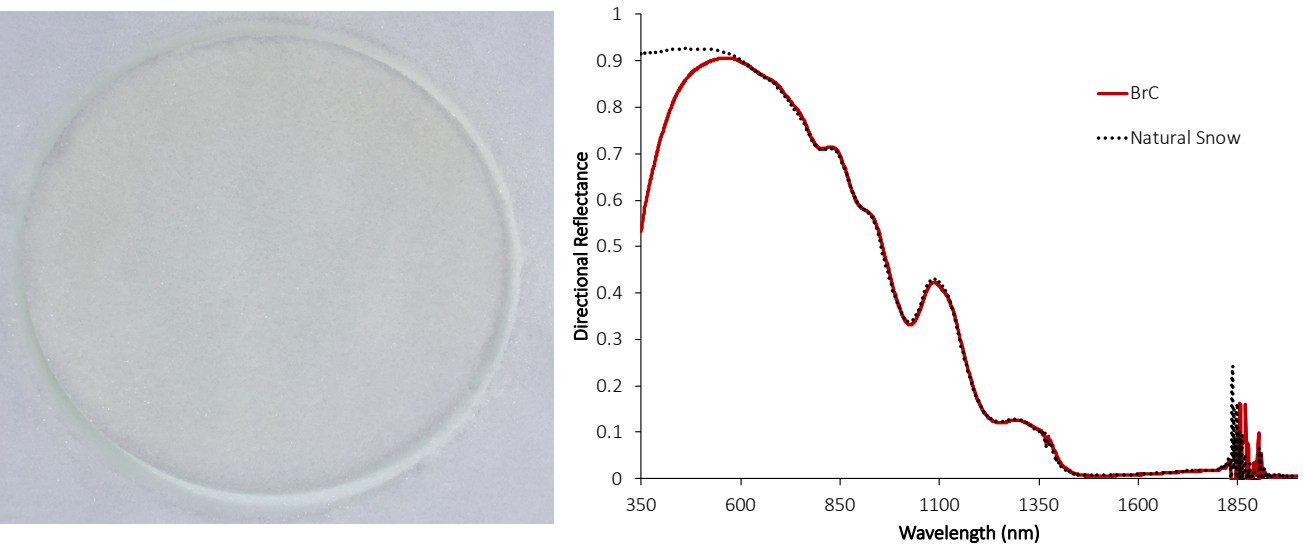

**Figure 6:** Image and HCRF spectra for a BrC deposition and adjacent natural snow in May of 2017 at the Tamarack Lake site. This BrC deposition drastically reduced the high natural snow reflectivity in the ultraviolet and short-wavelength visible (< 500 nm) spectral regions. Solar zenith: 34.6°; solar azimuth: 235.0°.

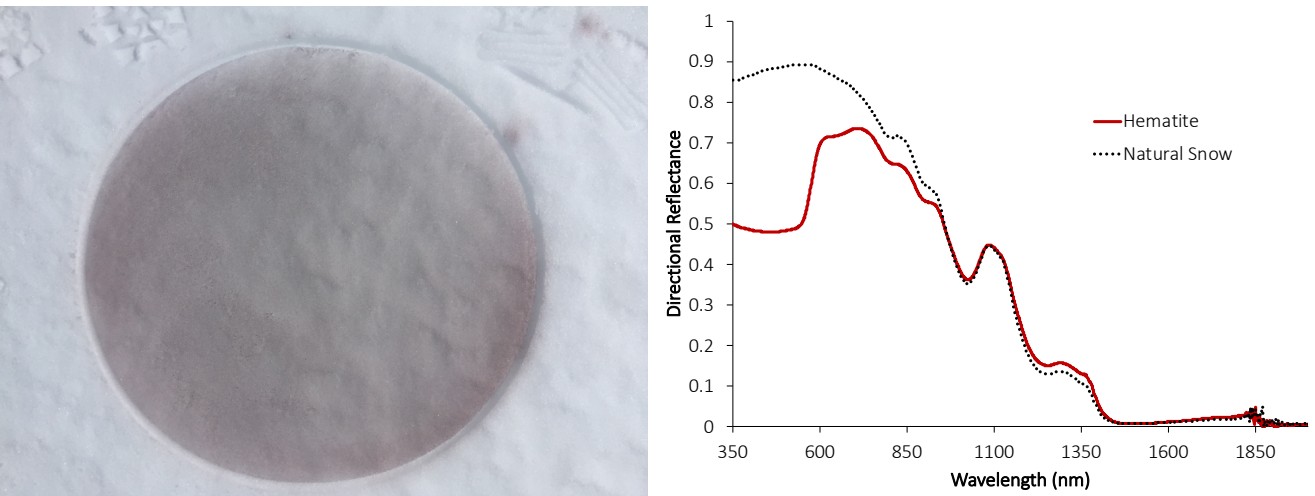

**Figure 7:** Image and HCRF spectra for a hematite deposition and adjacent natural snow in May of 2016 at the CUES research site. This hematite deposition drastically reduced the high natural snow reflectivity in the ultraviolet and short-wavelength visible (< 600 nm) spectral regions. Solar zenith: 26.7°; solar azimuth: 167.6°.