# Peer review of "Apparatus for Dry Deposition of Aerosols on Snow"

_Atmospheric Measurement Techniques, 2018_

## Referee Comment (RC1) · J. Dozier (Referee) · 21 Aug 2018

The manuscript describes customized instruments for experimentally depositing BC, BrC, and mineral dust on snow. The manuscript includes some cursory analysis of the results, based on measurements of the spectral HCRF (hemispherical-conical reflectance factor). However, given the wealth of the data gathered, this analysis could be more robust and help the reader appreciate the importance of the work.

Specifically:

The y-axes of Figures 4-7 are labeled "directional reflectance." The caption should include the geometry—solar zenith angle, observation angle and azimuth with respect to the sun. Although the "H" in HCRF designates "hemispherical," most of the illumination when skies are clear is in the direction from the sun.

The reflectance measurements (in the Supplement) should be sufficient to estimate the imaginary part of the refractive index. Skiles et al. (2016) have published a method to retrieve the imaginary part of the complex index of refraction from measurements of reflectance. It would be interesting to apply their method to these data. Moreover, Skiles' method could be compared to the published measurements for hematite (Scanza et al., 2015). Knowledge of the bulk optical properties of the absorbing particulates would be needed to model snow reflectance. Also important would be the size distribution, or at least the effective spherical radius, along with the particulate concentration in the upper snowpack. The shape of the spectral reflectance between the blue and red wavelengths depends partly on the size of the contaminating particles.

For the historical context on the experimental approach, the manuscript should address the work of Conway et al. (1996). Specifically, they examined the fate of deposited black carbon and volcanic ash, either remaining near the surface or washed downward during melt depending on size and composition. Sterle et al. (2013) also examined the fate of deposited BC, an important consideration in assessing the effect of absorbing aerosols on hydrology and climate.

A few nits: "Dozier" not "Dozer" page 1, line 23. "Sierra Nevada" means snowy mountain range, so eliminate "Mountains" on page 2, line 43. How about "Inside the" instead of "Inside of the" on page 3, line 9? In the Acknowledgments, you should include Ned Bair, who helped you a lot.

Conway, H., Gades, A., and Raymond, C. F.: Albedo of dirty snow during conditions of melt, Water Resources Research, 32, 1713-1718, doi 10.1029/96WR00712, 1996.

Scanza, R. A., Mahowald, N., Ghan, S., Zender, C. S., Kok, J. F., Liu, X., Zhang, Y., and Albani, S.: Modeling dust as component minerals in the Community Atmosphere Model: development of framework and impact on radiative forcing, Atmospheric Chemistry and Physics, 15, 537-561, doi 10.5194/acp-15-537-2015, 2015.

Skiles, S. M., Painter, T., and Okin, G. S.: A method to retrieve the spectral complex refractive index and single scattering optical properties of dust deposited in mountain snow, Journal of Glaciology, 63, 133-147, doi 10.1017/jog.2016.126, 2016.

Sterle, K. M., McConnell, J. R., Dozier, J., Edwards, R., and Flanner, M. G.: Retention and radiative forcing of black carbon in eastern Sierra Nevada snow, The Cryosphere, 7, 365-374, doi 10.5194/tc-7-365-2013, 2013.
* * *

---

## Referee Comment (RC2) · Anonymous Referee #2 · 29 Oct 2018

SUMMARY

The aim of the present manuscript is to describe a portable apparatus for the generation and deposition on snow of solid aerosol. The presentation of the method and results is logic, well organized and easy to read. Nevertheless, the apparatus is still under development and potentially affected by some biases. Moreover, the scarce amount of data limits the judgment of the apparatus performances: absolute concentration of deposited aerosol, temperature enhancement of air and snow in the deposition chamber. The discussion on the change of snow optical properties is very basic and based on the simple inverse proportionality between impurities presence and snow reflectance. Without knowing the amount of deposited aerosol, the overall meaning of such results is very limited. However the topic is of interest for the "snow community" and matches

the scientific domain of AMT, the lack of investigation of the above-mentioned issues makes me judge the submission of the manuscript as premature. I thus do not recommend the publication of the present work in its current status, and I encourage the authors to perform additional measurements and resubmit their results. The comments listed below might help the authors to improve their work.

MAJOR COMMENTS

1) Amount of deposited material As an apparatus for deposition, the range of aerosol in snow concentration that can be achieved should be known, this was unfortunately not quantified. Up to the reviewer, this lack affects the entire manuscript, limiting the assessment of deposition homogeneity and the understanding of radiative snow properties. First, the visual assessment of deposition is not exactly robust. From Figure 4 it is evident that there is a remarkable pattern of impurities dispersion, within the same experiment and among the different aerosol types. Second, in order to study radiative impact or potential migration of BC/BrC/dust, the operator should know the initial concentration of impurities. Here, without such information is extremely hard to contextualize the results shown in Figure 5, 6 and 7. The authors are encouraged to collect the snow and quantify the presence of particles across the deposition areas by nebulizing the snow with a pneumatic nebulizer (Lim et al., 2014) and by measuring the absorption coefficient (Ajtai et al., 2010) or refractory black carbon concentration (Katich et al., 2017). Preferably, the concentration of BC or absorption should be quantified before and after deposition.

2) Vertical penetration

While the vertical distribution of the impurities affects the overall light absorption through the snowpack (Tuzet et al., 2017), melting might change the vertical distribution of BC particles (Doherty et al., 2013). It is thus of extreme importance to know the exact location of the impurities layer. In the here presented setup, the particles are transported from the generator to the deposition chamber by an air stream. The

authors should verify if the air flow pushes the particles within the snowpack and to which depth the penetration occurs.

3) Temperature artifacts in the deposition chamber

The authors mentioned that the temperature in the chamber might increase during the deposition process and might modify the size and optical properties of the snow grains. However, this potential bias was not quantified. I suggest the authors repeating the deposition experiments without aerosol generation and simultaneously monitoring the air and snow temperature inside and outside the deposition chamber. Ideally, the assessment should be conducted under different environmental conditions: cloudy-sunny, cold-warm temperature. This method will provide an indication of the temperature increase inside the chamber. The subsequent and potential change in the properties of the snow such as liquid water content, density, specific surface area, and reflectance should be quantified. Beside the warming caused by the "greenhouse" effect of the chamber, I imagine that the exhaust of the combustion might contribute to the temperature enhancement. Although the long coil line (Figure 3) and the cold ambient temperatures might mitigate the heat transport, the potential warming effect should be assessed.

4) Combustion chamber

The in-situ generation of combustion generated particles is definitely interesting but of complex deployment, especially in extreme cold conditions (here tested at air temperature above 5°C), and might contaminate the surrounding snow (Figure 5 shows that deposition is not limited to the area below the deposition chamber). Moreover, the variety of fuels, the combustion efficiency (function of relative humidity, temperature, and altitude) limits the reproducibility of the experiments. The suspension of dry black carbon powder, similar to dust, might reduce the risk of contamination, increase the reproducibility of the experiments and reduce the weight of the entire apparatus. Did the author consider this option?

SPECIFIC COMMENTS

P3L22: the airflow exiting the combustion chamber is of approximately 5 L, where does the air exit the combustion chamber? At the bottom through the snow? Wouldn't this contaminate the surrounding snow? Did the authors ever consider the installation of an exhaust line with a total filter?

P3L30: do the authors know how much dust is actually transported to the deposition chamber?

P3L41: would a fan (moved by the airflow or a portable battery) enhance the dispersion of the particles?

F1-2: the schematics are basic, technical details should be added: airflow intake and output, size of combustion and deposition chambers, interested snow area.

REFERENCES

Ajtai, T., Filep, Á., Schnaiter, M., Linke, C., Vragel, M., Bozóki, Z., Szabó, G. and Leisner, T.: A novel multi−wavelength photoacoustic spectrometer for the measurement of the UV–vis-NIR spectral absorption coefficient of atmospheric aerosols, J. Aerosol Sci., 41(11), 1020–1029, doi:10.1016/j.jaerosci.2010.07.008, 2010. Doherty, S. J., Grenfell, T. C., Forsström, S., Hegg, D. L., Brandt, R. E. and Warren, S. G.: Observed vertical redistribution of black carbon and other insoluble light-absorbing particles in melting snow, J. Geophys. Res. Atmospheres, 118(11), 5553–5569, doi:10.1002/jgrd.50235, 2013. Katich, J. M., Perring, A. E. and Schwarz, J. P.: Optimized detection of particulates from liquid samples in the aerosol phase: Focus on black carbon, Aerosol Sci. Technol., 51(5), 543–553, doi:10.1080/02786826.2017.1280597, 2017. Lim, S., Faïn, X., Zanatta, M., Cozic, J., Jaffrezo, J.-L., Ginot, P. and Laj, P.: Refractory black carbon mass concentrations in snow and ice: method evaluation and inter-comparison with elemental carbon measurement, Atmos Meas Tech, 7(10), 3307–3324, doi:10.5194/amt-7-3307-2014, 2014. Tuzet, F., Dumont, M., Lafaysse, M., Picard, G., Arnaud, L., Voisin,

D., Lejeune, Y., Charrois, L., Nabat, P. and Morin, S.: A multilayer physically based snowpack model simulating direct and indirect radiative impacts of light-absorbing impurities in snow, The Cryosphere, 11(6), 2633–2653, doi:10.5194/tc-11-2633-2017, 2017.

---

## Author Comment (AC1) · 25 Nov 2018

The authors appreciate the detailed insights and suggestions by Dr. Dozier. We have considered all comments and have responded below.

**Jeff Dozier, reviewer #1**
**The manuscript describes customized instruments for experimentally depositing BC, BrC, and mineral dust on snow. The manuscript includes some cursory analysis of the results, based on measurements of the spectral HCRF (hemispherical-conical reflectance factor). However, given the wealth of the data gathered, this analysis could be more robust and help the reader appreciate the importance of the work.**

**Specifically:**
**The y-axes of Figures 4-7 are labeled "directional reflectance." The caption should include the geometry, solar zenith angle, observation angle and azimuth with respect to the sun.**

- The figures have been updated to reflect this information. Additionally, azimuth has been added to the table of information for each measurement (Table 1). The observation angle of the field spectrometer is nadir-looking for each measurement. This information has been added to the manuscript (p. 4, l. 33).

**Although the "H" in HCRF designates "hemispherical," most of the illumination when skies are clear is in the direction from the sun. The reflectance measurements (in the Supplement) should be sufficient to estimate the imaginary part of the refractive index. Skiles et al. (2016) have published a method to retrieve the imaginary part of the complex index of refraction from measurements of reflectance. It would be interesting to apply their method to these data. Moreover, Skiles' method could be compared to the published measurements for hematite (Scanza et al., 2015).**

- While the authors agree that methods outlined in Skiles et al. (2016) are useful, they are outside of the scope of this manuscript, which describes a deposition apparatus and method. However, it is a great example of a potential application of the apparatus described here and we therefore mention it in our manuscript as such (p. 6, l. 35-37).

**Knowledge of the bulk optical properties of the absorbing particulates would be needed to model snow reflectance. Also important would be the size distribution, or at least the effective spherical radius, along with the particulate concentration in the upper snowpack. The shape of the spectral reflectance between the blue and red wavelengths depends partly on the size of the contaminating particles.**

- We have updated the manuscript to include the particle size distributions for the three aerosols used in this particular study by placing this information in the supplemental data section. However, these size distribution data have been produced in laboratory settings and not in situ. For BrC and BC, the size distributions in the updated manuscript represent measurements made under nearly identical combustion conditions in laboratory settings during previous, related studies. For the hematite used in this study, we used a pure, artificially manufactured powder, for which a size distribution was measured by the manufacturer. For each of the three number size distributions presented, we supply the fractional lognormal distribution and the corresponding geometric mean diameter and geometric standard deviation.

- Future versions of the apparatus described in this manuscript could add ancillary characterization (e.g., size distribution) of the aerosols during deposition, and we have included this information in the updated manuscript (p. 6, l. 7-10).

**For the historical context on the experimental approach, the manuscript should address the work of Conway et al. (1996). Specifically, they examined the fate of deposited black carbon and volcanic ash, either remaining near the surface or washed downward during melt depending on size and composition. Sterle et al. (2013) also examined the fate of deposited BC, an important consideration in assessing the effect of absorbing aerosols on hydrology and climate.**

- The work presented in the manuscript represents depositions of aerosol that are in the topmost layer(s) of snow; an investigation to whether the aerosol migrate further into the snowpack is not included here. For each of the three aerosols used in the study, a new study site was chosen upwind of the last, as to not contaminate reflection measurements. Aerosols found in the topmost layer, as pointed out by the reviewer, may represent 1) aerosols that have been very recently deposited, such as a recent dust storm, or 2) aerosols that remain in the uppermost layer of snow during the ablation season when water soluble species are washed out with snowmelt. We thank the reviewer for pointing out yet another application of this apparatus and have added this to our section on potential applications in our manuscript (p. 6, l. 20-27).

**A few nits:**
- **"Dozier" not "Dozer" page 1, line 23.**

    o Fixed. Our apologies, Dr. Dozier.

- **"Sierra Nevada" means snowy mountain range, so eliminate "Mountains" on page 2, line 43.**
    o Fixed.

- **How about "Inside the" instead of "Inside of the" on page 3, line 9?**
    o Fixed.

- **In the Acknowledgments, you should include Ned Bair, who helped you a lot.**
    o Acknowledged.

---

## Author Comment (AC2) · 25 Nov 2018

The authors appreciate the critical but helpful comments of reviewer #2. He or she has pointed to a few key missing points to be made about the performance of deposition apparatus. However, the authors feel that after making some simple clarifications in the manuscript described below, we would encourage the reviewer to reconsider their judgement of "premature" and that this work can spark inspiration and innovation for very valuable experimental method of artificially depositing aerosol onto a snow surface to study aerosol-snow interactions for future researchers. We have outlined our responses in-line below and have updated the manuscript to reflex these changes.

**Anonymous reviewer #2**
**The aim of the present manuscript is to describe a portable apparatus for the generation and deposition on snow of solid aerosol. The presentation of the method and results is logic, well organized and easy to read. Nevertheless, the apparatus is still under development and potentially affected by some biases. Moreover, the scarce amount of data limits the judgment of the apparatus performances: absolute concentration of deposited aerosol, temperature enhancement of air and snow in the deposition chamber. The discussion on the change of snow optical properties is very basic and based on the simple inverse proportionality between impurities presence and snow reflectance. Without knowing the amount of deposited aerosol, the overall meaning of such results is very limited. However, the topic is of interest for the "snow community" and matches the scientific domain of AMT, the lack of investigation of the above-mentioned issues makes me judge the submission of the manuscript as premature. I thus do not recommend the publication of the present work in its current status, and I encourage the authors to perform additional measurements and resubmit their results. The comments listed below might help the authors to improve their work.**

**1) Amount of deposited material**
**As an apparatus for deposition, the range of aerosol in snow concentration that can be achieved should be known, this was unfortunately not quantified. Up to the reviewer, this lack affects the entire manuscript, limiting the assessment of deposition homogeneity and the understanding of radiative snow properties. First, the visual assessment of deposition is not exactly robust. From Figure 4 it is evident that there is a remarkable pattern of impurities dispersion, within the same experiment and among the different aerosol types. Second, in order to study radiative impact or potential migration of BC/BrC/dust, the operator should know the initial concentration of impurities. Here, without such information is extremely hard to contextualize the results shown in Figure 5, 6 and 7. The authors are encouraged to collect the snow and quantify the presence of particles across the deposition areas by nebulizing the snow with a pneumatic nebulizer (Lim et al., 2014) and by measuring the absorption coefficient (Ajtai et al., 2010) or refractory black carbon concentration (Katich et al., 2017). Preferably, the concentration of BC or absorption should be quantified before and after deposition.**

- To the reviewer's first point, Figure 4 represents the inhomogeneity of previous implementations of the apparatus (left and middle panel) to the current one (right panel). The caption of the image and the manuscript did not reflect this, and they have been updated appropriately.

- While the deposition uniformity is not quantified for this particular study, the authors point out that the current apparatus provides a sufficient uniformity for meaningful directional reflectance measurements of the deposited area and that of unaffected snow. While it may be

desirable for some applications to quantify the three-dimensional distribution of the deposited impurities in the snow, this is beyond the scope of our manuscript. It's important to note that other users may have very different uses for this experimental setup; the updated manuscript suggests some additional examples (Section 4). The present manuscript simply describes novel approaches of depositing aerosols onto the snow surface and thereby allowing for characterization of snow properties modified by such deposition. While, we use snow reflectance modified by the deposition of absorbing aerosols as an example, we have in no way attempted to characterize all properties or consequences of aerosol depositions.

-   Previous publications on artificially depositing or "doping" snow with light-absorbing impurities have shown that this is sometimes difficult, may be ineffective, and may require an overbearing amount of equipment. Here, we describe a simple apparatus for the deposition of both sub-micron combustion aerosols and super-micron mineral dust aerosols. A future study (Beres et al., in prep.) addresses the need to connect the albedo reduction with the mass of aerosol deposited.

**2) Vertical penetration**
**While the vertical distribution of the impurities affects the overall light absorption through the snowpack (Tuzet et al., 2017), melting might change the vertical distribution of BC particles (Doherty et al., 2013). It is thus of extreme importance to know the exact location of the impurities layer. In the here presented setup, the particles are transported from the generator to the deposition chamber by an air stream. The authors should verify if the air flow pushes the particles within the snowpack and to which depth the penetration occurs.**

-   The airflow is assumed to push aerosol onto and into the snowpack as there is no additional outlet or escape for airflow inside the deposition chamber present. This design may partly reproduce the phenomenon of air or wind pumping in which there is an exchange of air within and above the snowpack. Visually, we have examined a vertical profile of the aerosol deposited snow and found that the absorbing impurities are located nearly exclusively in the top 2 cm of the snowpack, where further uncertainties of their vertical location (i.e., within the top 2 cm) have very little influence on snowpack optics.
-   In the current manuscript, the verification of deposition by way of directional reflection measurements occurs immediately after the deposition; that is, the fate of the aerosol after deposition (whether any loss is due to melt scavenging, photochemical reactions, or other processes) is not considered. The authors (and others) are working to compile a manuscript that examines the fate of BrC aerosol deposited with the present apparatus and the depth at which they find deposited organic carbon, both before and after the experiment (Beres et al., 2019, in prep.).

**3) Temperature artifacts in the deposition chamber**
**The authors mentioned that the temperature in the chamber might increase during the deposition process and might modify the size and optical properties of the snow grains. However, this potential bias was not quantified. I suggest the authors repeating the deposition experiments without aerosol generation and simultaneously monitoring the air and snow temperature inside and outside the deposition chamber. Ideally, the assessment should be conducted under different environmental conditions: cloudy-sunny, cold-warm temperature. This method will provide an indication of the temperature increase inside the chamber. The subsequent and potential change in the properties of the snow such as liquid water content, density, specific surface area, and reflectance should be quantified. Beside the**

**warming caused by the "greenhouse" effect of the chamber, I imagine that the exhaust of the combustion might contribute to the temperature enhancement. Although the long coil line (Figure 3) and the cold ambient temperatures might mitigate the heat transport, the potential warming effect should be assessed.**

- We agree that temperature in the chamber is affected by outside air temperature, incoming solar radiation, outgoing thermal infrared radiation, and wind speed. However, for sufficiently short deposition periods, these effects can be neglected, but should be kept in mind for both experimental design and data analysis. A thorough quantification of these effects should be interesting but, given the extensive matrix of parameters, is outside the scope of our manuscript. However, one way to gauge the change in effective grain size is to employ the methods outlined by Nolin and Dozier (2000) in which they examine the 1.03 μm absorption feature present in snow reflectance spectra. The authors have performed this calculation and find that – for this particular set of depositions – there is no consistent increase in grain size when comparing the measured directional reflectance of the natural snowpack to that of the snow infused with absorbing aerosol. However, in response to the reviewer's comment, the authors have expanded the discussion portion of Section 3.3 "Challenges and further development" to include the reviewer's concerns and how they might be mitigated.

**4) Combustion chamber**
**The in-situ generation of combustion generated particles is definitely interesting but of complex deployment, especially in extreme cold conditions (here tested at air temperature above 5°C) and might contaminate the surrounding snow (Figure 5 shows that deposition is not limited to the area below the deposition chamber). Moreover, the variety of fuels, the combustion efficiency (function of relative humidity, temperature, and altitude) limits the reproducibility of the experiments. The suspension of dry black carbon powder, similar to dust, might reduce the risk of contamination, increase the reproducibility of the experiments and reduce the weight of the entire apparatus. Did the author consider this option?**

- The reviewer has noted some limitations of the deployment of this apparatus for in-situ deposition of combustion aerosol. The manuscript describes deposition of emissions of black and brown carbon produced on-site and this is part of the novelty of this device, which wasn't emphasized enough in the original manuscript. The authors are not aware of any similar methods to produce the albedo-reducing effect of combustion aerosols found in the atmosphere. However, there have been many attempts to reproduce this effect by using stand-ins such as Carbon Black or other dry BC-like powders. However, combustion aerosols are typically ~100 nm in diameter, and of fractal-like nature (Chakrabarty et al., 2006); such aerosols cannot easily be reproduced by de-agglomeration of powders due to the large adhesion forces found for sub-micron particles.

**SPECIFIC COMMENTS**
**P3L22: the airflow exiting the combustion chamber is of approximately 5 L, where does the air exit the combustion chamber? At the bottom through the snow? Wouldn't this contaminate the surrounding snow? Did the authors ever consider the installation of an exhaust line with a total filter?**

- The apparatus in its current form is designed to function in the very way that the reviewer is concerned about: by pumping aerosol *into* the snowpack. Indeed, the snowpack acts to filter

particulate matter (and other atmospheric constituents not discussed in this manuscript) due to air or wind pumping, a gas-exchange phenomenon between air in the snowpack and that of the atmosphere above (Kuhn, 2001). For our particular albedo-reducing verification (via directional reflectance measurements), contamination of the surrounding snow would only be an issue if the contamination was to the snow that we use as reference for "natural snow". The authors have taken care to not contaminate this snow by performing reference measurements more than 1 m away from deposition sites. For the current design of the apparatus, the installation of an exhaust line would deviate from the intended purpose of aerosol penetration into the snowpack.

**P3L30: do the authors know how much dust is actually transported to the deposition chamber?**

- No, the authors do not currently have a method to access the transport efficiency from the entrainment chamber to the snowpack when depositing mineral dust. For this study of the apparatus, the application of darkening the snow surface to better understand the impacts on snow surface albedo reduction due to absorbing aerosols was qualitative, and characterization of the transport efficiency for entrained material may be of future interest.

**P3L41: would a fan (moved by the airflow or a portable battery) enhance the dispersion of the particles?**

- A previous implementation of the apparatus described in the manuscript housed a single 12V DC fan at the approximate centroid of the deposition chamber, powered by portable battery. As the reviewer suggested, the authors hypothesized that a fan would help facilitate a uniform dispersion of particles throughout the deposition area. On the contrary, a fan caused the deposition to be much *less* uniform. In fact, the two leftmost panels of Figure 4 represent hematite deposition *with* a fan in use. It was only after removing the fan that hematite deposition became very uniform (far right panel of Figure 4 and Figure 7, left panel). The authors updated the manuscript to indicate the efficacy of previous iterations of development of the apparatus and the use of a fan in the production of depositions in Figure 4.

**F1-2: the schematics are basic, technical details should be added: airflow intake and output, size of combustion and deposition chambers, interested snow area.**

- The authors thank the reviewer for this valuable suggestion. Updated figures have been provided in the updated manuscript to include this information.

**References**

Beres, N. D., Sengupta, D., Samburova, V., Painter, T. H., and Moosmüller, H.: Brown Carbon on snow: reduction of spectral albedo and its implications, Atmos. Chem. Phys., manuscript in preparation, 2019.

Chakrabarty, R. K., Moosmüller, H., Garro, M. A., Arnott, W. P., Walker, J., Susott, R. A., Babbitt, R. E., Wold, C. E., Lincoln, E. N. and Hao, W. M.: Emissions from the laboratory combustion of widland fuels: Particle morphology and size, J. Geophys. Res. Atmos., 111(7), 1–16, doi:10.1029/2005JD006659, 2006.

Nolin, A. W. and Dozier, J.: A Hyperspectral Method for Remotely Sensing the Grain Size of Snow, Remote Sens. Environ., 74(2), 207–216, doi:10.1016/S0034-4257(00)00111-5, 2000.

Kuhn, M.: The nutrient cycle through snow and ice, a review, Aquat. Sci., 63(2), 150–167, doi:10.1007/PL00001348, 2001.